# CabZIP23 Integrates in CabZIP63–CaWRKY40 Cascade and Turns CabZIP63 on Mounting Pepper Immunity against *Ralstonia solanacearum* via Physical Interaction

**DOI:** 10.3390/ijms23052656

**Published:** 2022-02-28

**Authors:** Qiaoling Lu, Yu Huang, Hui Wang, Meiyun Wan, Jingang Lv, Xingge Cheng, Yuanhui Chen, Weiwei Cai, Sheng Yang, Lei Shen, Deyi Guan, Shuilin He

**Affiliations:** 1Key Laboratory of Applied Genetics of Universities in Fujian Province, Fujian Agriculture and Forestry University, Fuzhou 350002, China; 1190102021@fafu.edu.cn (Q.L.); 1190102014@fafu.edu.cn (Y.H.); 1200101011@fafu.edu.cn (H.W.); 1210102021@fafu.edu.cn (M.W.); 1210102016@fafu.edu.cn (J.L.); 1210102001@fafu.edu.cn (X.C.); y18950475671@163.com (Y.C.); 2170102016@fafu.edu.cn (S.Y.); 000q010036@fafu.edu.cn (D.G.); 2Agricultural College, Fujian Agriculture and Forestry University, Fuzhou 350002, China; 3College of Horticultural Sciences, Zhejiang A&F University, Hangzhou 311300, China; caivivi@zafu.edu.cn; 4College of Horticultural and Plant Protection, Yangzhou University, Yangzhou 225009, China; shenlei07@yzu.edu.cn

**Keywords:** *Capsicum annuum*, CabZIP23, CabZIP63, immunity, *Ralstonia solanacearum*

## Abstract

CabZIP63 and CaWRKY40 were previously found to be shared in the pepper defense response to high temperature stress (HTS) and to *Ralstonia solanacearum* inoculation (RSI), forming a transcriptional cascade. However, how they activate the two distinct defense responses is not fully understood. Herein, using a revised genetic approach, we functionally characterized CabZIP23 in the CabZIP63–CaWRKY40 cascade and its context specific pepper immunity activation against RSI by interaction with CabZIP63. CabZIP23 was originally found by immunoprecipitation-mass spectrometry to be an interacting protein of CabZIP63-GFP; it was upregulated by RSI and acted positively in pepper immunity against RSI by virus induced gene silencing in pepper plants, and transient overexpression in *Nicotiana benthamiana* plants. By chromatin immunoprecipitation (ChIP)-qPCR and electrophoresis mobility shift assay (EMSA), CabZIP23 was found to be directly regulated by CaWRKY40, and CabZIP63 was directly regulated by CabZIP23, forming a positive feedback loop. CabZIP23–CabZIP63 interaction was confirmed by co-immunoprecipitation (CoIP) and bimolecular fluorescent complimentary (BiFC) assays, which promoted CabZIP63 binding immunity related target genes, including CaPR1, CaNPR1 and CaWRKY40, thereby enhancing pepper immunity against RSI, but not affecting the expression of thermotolerance related CaHSP24. All these data appear to show that CabZIP23 integrates in the CabZIP63–CaWRKY40 cascade and the context specifically turns it on mounting pepper immunity against RSI.

## 1. Introduction

As sessile organisms, plants are frequently and inevitably exposed to attacks from pathogens and abiotic stresses such as high temperature stress. To survive, they have to defend themselves against these stresses, by mounting efficient defense responses upon the perception of the stress, via massive transcriptional reprogramming with the actions of various transcription factors [1,2,3]. Accumulating data indicate that the defense responses to different stresses should be coordinated, and the transcription factors involved in plant defense responses comprise transcription cascades or networks [4,5,6], providing the plants with great regulatory potential. However, how these transcription factors (TFs) cascades are organized and operate, in terms of plant coordinated response to different stresses, remains poorly understood

bZIP (basic leucine zipper) proteins, characterized by the conserved bZIP domain containing a basic region for DNA binding, and a more diversified leucine zipper region for protein dimerization [7], comprise a large gene family in plants [8,9,10,11,12,13]. The members of the bZIP family can be classified into 10 to 13 groups [14,15], by their targeting and binding cis-elements, such as the C and G-boxes in the promoters of their target genes [16,17,18]. To fulfill their function appropriately, bZIP proteins interact with themselves, or other bZIPs, as homodimers [19,20,21] or heterodimers [20,22,23,24,25,26], and form transcription cascades or networks with other TFs such as WRKY TFs, and in some cases, they are fine-tuned by other regulatory proteins via physical interaction [27,28,29]. bZIP proteins play roles in regulating multiple biological processes such as plant growth and development, as well as the plant immune response against pathogens [30,31,32], and the defense response to abiotic stresses including high stress [33,34,35,36]. For example, TGA (TGACGTCA cis-element-binding proteins) is one class of bZIP proteins that has been implicated in SA-dependent immunity; the functions of TGA TFs are modulated by NPR1, via nuclear translocation of NPR1 upon salicylic acid (SA) perception [2,37,38,39]. Upon heat stress, bZIP18 and bZIP52 in Arabidopsis (*Arabidopsis thaliana*) translocate from the cytoplasm to the the nuclei, where they jointly regulate the expression of genes related to heat stress tolerance [40]. Another well characterized example is HY5, which has been implicated in a spectrum of biological processes including growth, development, photomorphogenesis [41,42], and in the response to abiotic stresses including salinity [43] and UV-B [44], as well as playing a role in host plant defense [45]. The function of HY5 appears to be modulated by other different regulatory proteins in a context specific manner; it interacts with RSM1, an MYB protein, to regulate ABI5 [43], it is regulated by WRKY36 and UVR8 [46], associates with E3 ubiquitin ligase COP1 [47], and directly regulates R2R3-MYB transcription factor gene MYB12, that is a key activator of flavonol biosynthesis [48], while upon attack of pathogens, HY5 regulates the expression of ENHANCED DISEASE SUSCEPTIBILITY 1 (EDS1), by directly binding to its promoter, resulting in red-light induced programmed cell death [49]. All these results indicate that some bZIPs may play important roles in the coordination of plant response to multiple stresses. Even though bZIPs have been found to play roles in plant immunity and heat stress response, whether and how they operate in coordinating plant immunity and heat stress response remain to be elucidated.

Pepper (*Capsicum annuum*) is a kind of solanaceous vegetables of great agricultural importance worldwide; it originated in tropical and subtropical regions of Central and South America and is planted in the warm seasons in these regions, or in greenhouse all over the world. Like other Solanaceaes, such as tomato (*Solanum lycopersicum*) and tobacco (*Nicotiana tabacum*), pepper is frequently exposed to attacks from soil-borne pathogens such as *Ralstonia solanacearum* (Smith 1896), infection (RSI), and challenges from high temperature stress (HTS), which lead to impairment in its growth and development, and heavy loss in its production. These stresses might have exerted a clear influence on the evolution and, therefore, the traits of pepper and other Solanaceaes. Our previous studies have demonstrated that CabZIP63 [36], CaWRKY6 [50], CaWRKY40 [51], CaNAC2c [1] and CaCDPK15 [52] are shared in pepper defense response to RSI and to HTS, and among these components, CaWRKY40 is transcriptionally and positively regulated by, and forms a feedback loop with, CabZIP63. However, how CabZIP63 is feedback modulated by CaWRKY40, and how these TFs function appropriately in pepper response to a given stress, remain poorly understood.

In the present study, we found that CabZIP23 participates in a CabZIP63–CaWRKY40 cascade, with CabZIP23 being regulated by CaWRKY40 and CabZIP63 being directly regulated by CabZIP23. CabZIP23 was regulated by RSI and acted positively in pepper immunity against RSI by interaction with CabZIP63, thereby promoting the targeting and regulation of immunity related genes including CaPR1, CaNPR1 and CaWRKY40, but did not affect the expression of CaHSP24 and thermotolerance.

## 2. Results

### 2.1. CabZIP23 Was Present in Proteins Isolated from CabZIP63-GFP Transiently Overexpressed in Pepper Leaves by Co-Immunoprecipitation Using an Antibody of GFP

Our previous study showed that CabZIP63 acts as a positive regulator in pepper response to RSI and HTS [53]. We speculate that it might fulfill its function by interaction with other proteins to activate an immunity or thermotolerance context specifically. To isolate its interacting partners, CabZIP63-GFP was transiently overexpressed in pepper leaves by agroinfiltration. The total proteins were isolated and the proteins that might interact with CabZIP63 were co-immunoprecipitated, using an antibody of GFP. The acquired proteins were subjected to MS assay, and a total of 82 proteins were identified. Among these proteins, a protein containing a conserved BRLZ domain and an NLS in its N’ terminal, and sharing a high homology sequence with bZIP23 in other plant species, such as the tomato, aroused our interest (Appendix A), since bZIP proteins have been previously found to perform their function as heterodimers by interacting with other bZIP proteins [20,22,23,24,25,26]. We named it CabZIP23, and assayed the cis-elements in its promoter region; a subset of well documented cis-elements related to plant immunity or defense response to abiotic stresses, such as DRE-[54], TGACG-[18], TGA-[55] and W-box [56], were found, implying that CabZIP23 might be involved in pepper immunity or its response to other abiotic stresses (Figure 1a). To test this speculation, we assayed the transcript levels of CabZIP23 in pepper plants challenged with RSI or HTS, as well as in response to the exogenous application of hormones or regulators, such as the application of abscisic acid (ABA), indole-3-acetic acid (IAA), trans-Zeatin (tZ), methyl jasmonate (MeJA) or SA. We found that CabZIP23 was upregulated by RSI but not by HTS, and was induced by exogenously applied SA or MeJA, but not by the treatment of exogenously applied ABA, IAA or tZ (Figure 1b,c). These results indicate that CabZIP23 might be involved in pepper immunity against RSI and is associated with immune signaling mediated by SA and JA [57,58].

### 2.2. CabZIP23 Is a Nuclear Protein and Has Transcriptional Activity

As a putative bZIP transcription factor, CabZIP23 might be located in the nucleus. To test this possibility, we assayed the subcellular localization of CabZIP23-GFP by agroinfiltration in the leaves of *Nicotiana benthamiana*. The results showed that GFP in the epidermal cells of CabZIP23-GFP agroinfiltrated in *N. benthamiana* leaves, were exclusively found in the nucleus, in agreement with H2B–RFP, which target to the nuclei (Figure 2a), while that in the GFP agroinfiltrated *N. benthamiana* leaves was found all over the cell. This result indicated that CabZIP23 is a nuclear protein. To assay whether CabZIP23 acts as a transcription factor, its transcriptional activity was assayed in yeast. To do this, the full length of the ORF of CabZIP23 (using the transcription factor CabZIP63 as a positive regulator) was fused with a GAL4 DNA-binding domain in the yeast expression vector pGBPKT7. The empty pGBPKT7 and the bait vector were transformed into the yeast strain Y2H Gold and incubated on SD/-Trp/X-α-gal or in SD/-Trp/-His medium. As shown in Figure 2b, yeasts harboring pGBPKT7-CabZIP23, pGBPKT7-CabZIP63, and the empty pGBPKT7, all grew on SD/-Trp medium, however, only yeasts harboring pGBPKT7-CabZIP23 and pGBPKT7-CabZIP63 grew in SD/-Trp/-His medium. The yeasts harboring pGBPKT7-CabZIP23 and pGBPKT7-CabZIP63 showed significant β-galactosidase activity after the addition of X-α-gal (Figure 2b), indicating that CabZIP23 exhibits transcriptional activity.

### 2.3. The Silencing of CabZIP23 Increased Susceptibility of Pepper Plants to RSI but Did Not Affect Pepper Thermotolerance

To test whether CabZIP23 plays a role in pepper immunity against RSI, we employed a loss-of-function assay to study the effect of CabZIP23 silencing by virus induced gene silencing (VIGS). To avoid the silencing of other members in the family of bZIP proteins, we used a fragment within 3’UTR of about 300 bps in length to construct the vector for the VIGS assay, and the silencing efficiency and specificity were tested by measuring the transcript levels of CabZIP23 in CabZIP23 silenced pepper plants challenged by RSI. The results showed that the transcript levels of CabZIP23 in CabZIP23 silenced pepper plants challenged by RSI was less than 10% of that in the mock treated wild type plants (Figure 3a). Using the CabZIP23 silenced pepper plants, we found that the silencing of CabZIP23 significantly increased the susceptibility of the pepper plants to RSI, displayed by more heavy wilt symptoms in the CabZIP23 silenced pepper plants than that in the mock treatment (Figure 3b). Consistently, the CabZIP23 silenced pepper plants supported a higher level of RS growth (Figure 3c), and higher level of disease index (Figure 3d). Consistently, the upregulation of immunity related CabZIP63, CaNPR1, CaPR1 and CaDEF1 by RSI was significantly blocked by CabZIP23 silencing (Figure 3e). By contrast, the silencing of CabZIP23 did not affect either pepper thermotolerance or the expression of thermotolerance related CaHSP24 (Appendix A). All these data indicate that CabZIP23 acts as a positive regulator in pepper immunity against RSI but not in thermotolerance.

### 2.4. The Transient Overexpression of CabZIP23 Triggered Cell Death and Upregulated Immunity Related Marker Genes

To confirm the result that CabZIP23 acts as a positive regulator in pepper immunity against RSI by VIGS, we performed the transient overexpression of CabZIP23-GFP by agroinfiltration to study its effect on HR mimicked cell death and the expression of immunity related genes. To do this, *Agrobacterium tumefaciens* GV3101 cells containing 35S::CabZIP23-GFP (using 35S::*GFP* as a control) were infiltrated into the leaves of pepper plants, and the success of the transient overexpression of CabZIP23-GFP was confirmed by quantitative real-time PCR (qRT-PCR) at 24 or 48 h post-infiltration (hpi), and western blotting using an antibody of GFP at 48 hpi (Figure 4a,b). We found that the transient overexpression of CabZIP23 triggered a clear hypersensitivity reaction (HR), mimicked cell death displayed by a higher level of ion leakage, manifested by conductivity and darker trypan blue staining, as well as a higher level of H_2_O_2_ accumulation, that was displayed by darker diaminobenzidine (DAB) staining (Figure 4c,d). In addition, the tested immunity related marker genes including CabZIP63, CaNPR1, CaPR1, and CaDEF1 were all induced at transcriptional level by the transient overexpression of CabZIP23 (Figure 4e). All these data supported the result that CabZIP23 acts positively in pepper immunity against RSI.

### 2.5. CabZIP23 Targeted the Promoters of CaPR1 and CaNPR1 in a G-Box Dependent Manner but Not That of CaHSP24

Since G-boxes have been frequently found to have the ability to be bound by bZIP proteins [16,17,18], and G-boxes were found to be present in the promoters of CaNPR1, which was found to be upregulated by the transient overexpression of CabZIP23, we, therefore, tested whether these genes can be directly targeted by CabZIP23, by an EMSA using prokaryotic expressed CabZIP23-GST and Cy5 fluorescence marked wild type probes (promoter fragments of the tested genes), as well as fluorescence unlabeled cold probes. A clear binding signal (mobility shift) was detected in the promoters of CaPR1 and CaNPR1 (G-box1 and G-box2), while no binding signal was detected in the promoter of thermotolerance related CaHSP24 (Appendix A). These data indicate that CabZIP23 might activate the expression of the immunity related genes, at least partially, by direct targeting. 

### 2.6. CabZIP23 Was Targeted and Transcriptionally Regulated by CaWRKY40

Since CabZIP23 was originally found to probably interact with CabZIP63, which is functionally and expressionally related to CaWRKY40, by our previous study [36], and a W-box is present in the promoter of CabZIP23, these results appear to suggest that CabZIP63 might be directly targeted by CaWRKY40. To test this possibility, we performed an immunoprecipitate from fragmented chromatins isolated from CaWRKY40-GFP, transiently overexpressed in the leaves of pepper plants challenged with RSI, and a specific pair of primers of W-box containing promoter fragments of CabZIP23. The result showed a clear enrichment of CaWRKY40 in the W-box containing CabZIP23 promoter (Figure 5a), indicating that CabZIP23 might be directly targeted by CaWRKY40. To test whether CabZIP23 was transcriptionally regulated by CaWRKY40, we measured the transcript level of CabZIP23 in CaWRKY40 transiently overexpressed pepper leaves and found that CabZIP23 was significantly upregulated by CaWRKY40 transient overexpression (Figure 5b). All these data indicate that CabZIP23 is positively and directly regulated by CaWRKY40.

### 2.7. CabZIP63 Was Directly Targeted and Transcriptionally Regulated by CabZIP23

Since C/G-box is present in the promoter of CabZIP63, which was positively feedback regulated by CaWRKY40, we speculate that CabZIP63 might be directly regulated by CabZIP23, through which CaWRKY40 regulates CabZIP63. To test this hypothesis, we performed a ChIP-qPCR to test whether the C/G-box containing promoter of CabZIP63 is bound by CabZIP23; the result showed a clear enrichment of CabZIP23 in the promoter of CabZIP63 (Figure 6a). We further studied the effect of CabZIP23 transient overexpression or its silencing on the transcription of CabZIP63. The results showed that the transient overexpression of CabZIP23 significantly upregulated CabZIP63, while its silencing significantly blocked the upregulation of CabZIP63 by RSI in pepper plants (Figure 6b,c). By contrast, the silencing of CabZIP23 significantly reduced the upregulation of CabZIP63 and CaWRKY40 by RSI, but not by HTS. All these results indicate that CabZIP63 is directly and positively regulated, and CaWRKY40 is indirectly regulated by CabZIP23 during pepper response to RSI, since CaWRKY40 is directly and positively regulated by CabZIP63 upon RSI [36]. 

### 2.8. The Confirmation of CabZIP23–CabZIP63 Interaction

To confirm the interaction between CabZIP23 and CabZIP63, we first performed a bimolecular fluorescent complimentary (BiFC) assay in the leaves of *N*. *benthamiana*. To conduct this, the N- and C-terminal portions of YFP were fused to CabZIP23 and CabZIP63, to generate CabZIP23-YFP^N^ and CabZIP63-YFP^C^, respectively. The interactions between the fusion proteins were visualized in the epidermal cells in the leaves of *N*. *benthamiana* at 48 hpi, via an Agrobacterium mediated transient expression system; the YFP signal was observed exclusively in the nuclei, as displayed by the H2B–RFP. To further confirm this result, we performed a co-immunoprecipitation (Co-IP) assay using the total protein isolated from CabZIP23-Myc and CabZIP63-Flag co-transiently overexpressing pepper leaves. The protein was first incubated with an antibody of Myc to immunoprecipitate CabZIP23-Myc, and the presence of CabZIP63 was detected by immunolotting with an antibody of Flag. The result showed that CabZIP23 interacts with CabZIP63 in vivo (Figure 7). These data indicate that CabZIP23 interacts with CabZIP63 in the nucleus. 

### 2.9. CabZIP23 Promoted the Binding of CabZIP23 to the Promoters of CaPR1 and CaNPR1 as Well as CaWRKY40, thereby Promoting Their Activations by CabZIP63

To study the effect of CabZIP23–CabZIP63 on pepper immunity, HR mimicked cell death triggered by the transient overexpression of CabZIP63 in TRV::CabZIP23 pepper leaves was performed by agroinfiltration. To do this, *A*. *tumefaciens* GV3101 cells containing 35S::CabZIP63-GFP were infiltrated into the leaves of TRV::CabZIP23 pepper plants. HR mimicked cell death, displayed by trypan blue staining, and H_2_O_2_ accumulation, displayed by DAB staining, were measured at 48 hpi. The results showed that the transient overexpression of CabZIP63 triggered cell death and a high level of H_2_O_2_ accumulation, which was blocked in the leaves of TRV::CabZIP23 pepper plants, indicating that CabZIP23 is required in the immunity mediated by CabZIP63 (Figure 8a). Consistently, we also found that the transient overexpression of CabZIP63 triggered upregulation of CaPR1 and CaNPR1 but did not affect the expression of CaHSP24 (Figure 8b). To test whether the effect of CabZIP23 silencing on HR mimicked cell death, mediated by CabZIP63, is associated with the modulation of CabZIP63 targeting by CabZIP23, we performed an EMSA to test the effect of the presence of CabZIP23 on the binding of CabZIP63 to the promoters of tested immunity or thermotolerance related marker genes, including CaPR1 and CaNPR1, which are targets of CabZIP63 during pepper immunity against RSI. The result showed that the excess application of CabZIP23 promoted the binding of CabZIP63 to the tested promoters of immunity related genes, and the presence of CabZIP23 led to the presence of heterodimers or heteromultimers. Noticeably, CabZIP63 slightly bound the promoter of CaHSP24, and the addition of CabZIP23 reduced the binding of CabZIP63 to the promoter of CaHSP24 (Figure 8c). To study the effect of CabZIP23 on the regulation of the three target genes by CabZIP63 in vivo, the DNA derived from CabZIP63-GFP transiently overexpressed and CabZIP23-silenced pepper leaves was isolated and used as a template for qRT-PCR. The results showed that the silencing of CabZIP23 almost completely blocked the enrichment of CabZIP63 on the promoter of the tested immunity related genes but not on the promoter of CaHSP24 (Figure 8d). In addition, we also tested the effect of CabZIP23 on the regulation of CaWRKY40 expression by CabZIP63; we obtained the similar result that the silencing of CabZIP23 significantly blocked the binding of CabZIP63 to the promoter of CaWRKY40 and, therefore, reduced its expression level (Appendix A). All these results indicate that CabZIP23 promotes the activation of immunity related genes and HR mimicked cell death by CabZIP63.

## 3. Discussion

The findings in our previous studies indicate that pepper, and probably other Solanacaes such as tobacco and tomato, employ shared mechanisms such as TFs including CabZIP63 and CaWRKY40, to cope with attack from RSI and HTS. However, how they functionally associate with each other and how they activate distinct context specific defense reactions remain unclear. The results in the present study reveal that CabZIP23 acts positively in pepper immunity against RSI, participates in the feedback loop between CabZIP63 and CaWRKY40, and enables CabZIP63 to specifically activate pepper immunity.

### 3.1. CabZIP23 Acts Positively in Pepper Response to RSI

Our data showed that when pepper plants were challenged by RSI, CabZIP23 was upregulated at the transcription level, and its silencing enhanced the susceptibility of pepper plants to RSI, coupled with the downregulation of pathogenesis-related (PR) genes including CaPR1, CaNPR1 and CaDEF1, which are shown to be immunity related by previous studies [59,60,61,62,63,64]. The clear phenotypic effect of the silencing of CabZIP23 indicates that it acts non-redundantly and positively in pepper response to RSI, supporting the notion that genes are transcriptionally modified in plant response to pathogens [50,51,65]. CabZIP23 was found to probably interact with CabZIP63, a bZIP TF that targets C/G-box and acts positively in pepper immunity against RSI and thermotolerance [36], however, unlike CabZIP63 that upregulates upon RSI and also upon HTS, CabZIP23 was not upregulated by HTS treatment (Figure 1b). Consistently, the silencing of CabZIP23 did not affect either the basal thermotolerance of pepper plants or the expression of thermotolerance related CaHSP24. These data indicate that CabZIP23 is not involved in the pepper defense response to HTS.

### 3.2. CabZIP23 Integrates into the Feedback Loop between CaWRKY40 and CabZIP63

It has been generally found that a subset of TFs has been activated at transcriptional level in reaction to pathogens or abiotic stresses [15,66,67,68,69,70]. These transcription factors are functionally connected, forming transcriptional cascades or networks [2,4,6,66,71]. Another phenomenon in the regulation of plant response to pathogen attack or to abiotic stress exposure, is the existence of positive or negative feedback loops [52,72,73,74]; this hierarchical organization of multiple TFs and feedback loops might benefit the plants in flexibly and appropriately regulating their defense response [6]. Our previous studies indicate that CaWRKY40 and CabZIP63 phenocopied each other in positively regulating pepper response to RSI and to HTS, with CaWRKY40 being directly targeted by CabZIP63; CabZIP63 and CaWRKY40 form a positive feedback loop during the pepper response to RSI and to HTS [36,51]. However, the promoter of CabZIP63 is not bound by CaWRKY40, indicating that some other components might exist between CaWRKY40 and CabZIP63 [36,51]. The data in the present study reveal that CabZIP23 was bound and positively regulated at the transcriptional level by CaWRKY40, while CabZIP63 was directly and transcriptionally regulated by CabZIP23 (Figure 6). Since CabZIP63 is modulated by a CDPK protein via phosphorylation (our unpublished data), it can be speculated that CabZIP63 locates upstream of CaWRKY40 and CabZIP23 is probably located in the feedback loop between CabZIP63 and CaWRKY40 during the pepper immune response to RSI.

### 3.3. CabZIP23 Promotes CabZIP63 to Context Specifically Mount Immunity against RSI via Physical Interaction

Given that CabZIP63 is shared by the pepper defense response to HTS and to RSI [75], it is crucial for pepper to employ this shared TF to appropriately activate thermotolerance and immunity, the two distinct defense responses [76,77,78,79], as for example, the accumulation of ROS such as H_2_O_2_ which are scavenged during thermotolerance, are triggered during the immune response. The data in the present study indicate that CabZIP63 was promoted to activate immunity against RSI by CabZIP23, which itself is upregulated by RSI. Like AtbZIP16 and AtbZIP68 in Arabidopsis [26], CabZIP23 interacts with CabZIP63 forming heterodimers, and this interaction promotes CabZIP63 binding to the promoter of the tested immunity related genes, including CaPR1, CaNPR1 and CaWRKY40, in a G-box dependent manner, thereby enhancing their expressions (Figure 7, Figure 8 and Appendix A, amplifying immune signaling via the CabZIP23–CabZIP63–CaWRKY40 feedback loop, and enhancing pepper immunity against RSI. Similarly, CaASR1 was found previously to also promote CabZIP63 to activate pepper immunity against RSI [75], indicating that CabZIP63 might be regulated at post-translational level by multiple regulatory proteins in specifically activating pepper immunity. Our previous studies also indicate that CaWRKY40 and CaNAC2c are also shared in pepper immunity against RSI and in thermotolerance, and that the function of CaWRKY40 and CaNAC2c in positively regulating pepper immunity was also found to be activated upon RSI by CaWRKY28 and CaNAC029, respectively, via physical interaction [1,51]. However, the role of CaNAC2c as a positive regulator in pepper thermotolerance is activated by CaHSP70 in a context specific manner [1], implying that the function of CabZIP63 as a positive regulator in pepper thermotolerance might be modulated at translational level by some unidentified regulatory proteins. Given that pepper employs shared signaling components including CaCDPK15 [52], CabZIP23, CabZIP63 [36], CaWRKY6 [50], CaWRKY40 [51], and CaNAC2c [1] to activate pepper immunity against RSI and thermotolerance, we speculate that a set of shared signaling components combined with precise and swift translational regulation might be a general and efficient strategy to adapt to different stresses that occur frequently. Therefore, to improve bacterial wilt resistance in pepper, these shared components at transcriptional level, and specific regulators of post-translational regulation, must be modified simultaneously and in a coordinated manner.

Conclusions: CabZIP23 is upregulated by RSI and acts positively in pepper immunity against RSI. It transcriptionally and directly regulated CabZIP63, and interacts with CabZIP63 forming heterodimers; in this way, CabZIP63 targeting and the transcriptional regulation of its immunity related target genes was thereby enhanced by CabZIP23.

## 4. Materials and Methods

### 4.1. Plant Materials and Growth Conditions

Seeds of HN42, a pepper (*C*. *annuum*) inbred line with a middle level of thermotolerance and bacterial wilt resistance, and *Nicotiana benthamiana* (Benjamin Bynoe), provided by the pepper breeding group in the Fujian Agriculture and Forestry University (Fuzhou, China), were sown in a soil mix (peat moss: perlite, 2:1 (*v*/*v*)) in plastic pots and placed in a growth room under the conditions of 25 °C, a relative humidity of 70%, 60–70 µmol photons m^−2^ s^−1^, and a 16 h light, 8 h dark photoperiod.

### 4.2. Pathogens and Inoculation Procedures

*Ralstonia solanacearum* strain FJC100301 was isolated previously in our lab and amplified according to the method of Dang [79]. The bacterial cell solution used for the RSI of the pepper plants for functional characterization of CabZIP23 was diluted to 10^8^ cfu mL^−1^ (OD_600_ = 0.8) with 10 mM MgCl_2_. For the RSI in the pepper leaves, 0.5 mL of the resulting *R*. *solanacearum* suspension was injected into the third pepper leaves from the eight-leaf stage from the apical meristem, by utilizing a syringe without a needle, and the mock was injected with sterile 10 mM MgCl_2_. The leaves were harvested at the indicated time points for the preparation of RNA or for other assays. When the pepper plant TRV::CabZIP23 and TRV::*00* grew to the stage of 6–8 leaves, the roots were wounded by scissors and then, using 5 mL of *R*. *solanacearum* strain FJC100301, irrigation of the injured roots was conducted. Then the pots were kept in a growth room at 28 °C with soil moisture at >90%. The disease indexes of the plants were evaluated at 15 days post-inoculation (dpi).

### 4.3. Vector Construction

For vector construction, a Gateway cloning technique (Invitrogen, Carlsbad, CA, USA) and a series of Gateway-compatible destination vectors were employed. The full-length cDNA of CabZIP23 was amplified and cloned into the entry vector pDONR207 by BP reaction, and then into the destination vectors such as pDEST15 and pEarleyGate103 by LR reaction for transient overexpression, subcellular localization, ChIP and EMSA analysis, respectively.

For CabZIP23 silencing analysis, the CDS of CabZIP23 was used for VIGS vector construction; its sequence specificity was confirmed by genome-wide homology sequence searching by BLAST against sequences in the CM334 and Zunla-1 databases. The specific DNA fragment of CabZIP23 was amplified by PCR with the specific primer pairs using cDNA of Zunla-1 as a template, and then cloned into pDONR207 by BP reaction. After confirmation by sequencing, they were further cloned into the pYL279 (TRV2) vector by LR reaction. All of the vectors were transformed into *A. tumefaciens* strain GV3101.

### 4.4. VIGS

The resulting Tobacco rattle virus (TRV) based vectors TRV2::CabZIP23 and TRV1 were transformed into *A. tumefaciens* strain GV3101. The GV3101 cells harboring TRV1 and TRV2::*00* were regarded as a negative control (resuspended by induction medium and mixed at a 1:1 ratio, OD_600_ = 0.8), and were co-infiltrated into cotyledons of 2-week-old pepper plants. The details of the process were as described in our previous studies [80].

### 4.5. Transient Overexpression in Pepper Leaves

For transient overexpression analysis, *A. tumefaciens* strain GV3101 cells harboring the 35S::CabZIP23 or 35S::*00* vector were grown overnight in LB medium, and then resuspended in induction medium (10 mM MES, 10 mM MgCl_2_, pH 5.4, and 150 µM acetosyringone). The bacterial suspension (OD_600_ = 0.8) was injected into the leaves of pepper plants at the eight-leaf stage, and the infiltrated 4 pepper leaves (4 biological repeats) were harvested at 24 or 48 h post-infiltration (hpi) for further assays.

### 4.6. Subcellular Localization of CabZIP23

The bacterial suspension of *A. tumefaciens* strain GV3101 containing the constructs *35S*::CabZIP23-GFP or *35S::GFP* were injected into *N. benthamiana* leaves using a syringe without a needle. At 48 hpi, using a laser scanning confocal microscope (TCS SP8, Leica, Solms, Germany) green fluorescent protein (GFP) signaling was imaged, with an excitation wavelength of 488 nm and a 505–530 nm band-pass emission filter. This experiment uses one sample and one biological repeat.

### 4.7. Yeast Transcriptional Activation

The pGBKT7 vectors containing the open reading frame (ORF) of CabZIP23 were transferred into yeast Y2H Gold competent cells. The bacteria solution was added to petri dishes lacking SD/-Trp amino acids and petri dishes lacking SD/-Trp/-His amino acids, and the concentration gradient was 10^0^, 10^−1^, 10^−2^. They were observed for two days in an incubator at 28 °C.

### 4.8. Measurement of Ion Conductivity

After the transient overexpression of CabZIP23, the leaves were collected, and six round leaves with a diameter of about 5 mm were each punched with a hole drill. The round leaves were immersed into a centrifugal tube filled with 10 mL ddH_2_O, and the leaves were shaken on a shaking table at low speed for about 1 h to measure the electrical conductivity by a Mettler Toledo 326 ion meter. This experiment uses 6 samples for 6 biological repeats.

### 4.9. Chlorophyll Fluorescence Spectrophotometry

For the thermotolerance assay, using leaves of TRV::CabZIP23 and TRV::*00* pepper plants, we treated the material in an illumination incubator with the conditions of 37 °C, 90% relative humidity, and a 16 h light, 8 h dark photoperiod for 48 h. According to the method of Schreiber, the pepper plants were adapted to the darkness for 15 min before being placed into the MINI Imaging PAM instrument (Heinz Walz GmbH, Effeltrich, Germany) to measure Fv/Fm and ∆F/Fm′ values from the pepper leaves [80]. This experiment uses 3 samples for 3 biological repeats.

### 4.10. Histochemical Staining

To assess the accumulation of H_2_O_2_ and reactive oxygen species, staining with trypan blue and DAB was carried out, according to the previously published method of Choi [81].

### 4.11. ChIP Analysis

ChIP assays were performed according to a previous study [82]. We inoculated pepper leaves with *A. tumefaciens* strain GV3101 cells harboring the 35S::CabZIP23-*GFP* construct, the infiltrated leaves were harvested at 48 hpi, ground into a powder under liquid nitrogen and crosslinked in a 1% formaldehyde solution. Using a probe sonicator, chromatin was isolated and sheared into 300–500 bp fragments, followed by the immunoprecipitation of the DNA–protein complexes using anti-GFP antibodies. This experiment uses 3 samples for 3 biological repeats.

### 4.12. Prokaryotic Expression of Fusion Protein in Escherichia coli

To obtain plenty of soluble CabZIP23 protein, pDEST-15 (tag GST) or pDEST-17 (tag 6×His) plasmids harboring the full-length ORF of CabZIP23 were transformed into the *Escherichia coli* (*E. coli*) strain BL21 (DE3). The expression of the fusion protein was induced using isopropyl-β-D-1-thiogalactopyranoside (IPTG; 1 mM) at 20 °C for 12 h, and an SDS-PAGE assay was performed to confirm whether the soluble fusion protein was present in the supernatant of the *E. coli* cell lysate.

### 4.13. Electrophoresis Mobility Shift Assay

To obtain a large number of pure target proteins and confirm the binding of CabZIP63 or CabZIP23 to the cis-elements within the probe, prokaryotic expression and an electrophoresis mobility shift assay (EMSA) were performed, and the methods were as described in our previous report [83].

### 4.14. Bimolecular Fluorescent Complimentary (BiFC) Assay

*N. benthamiana* leaves were infiltrated with 35S::CabZIP23-YFP^C^ and 35S::CabZIP63-YFP^N^ and harvested at 48 hpi. Yellow fluorescent protein (YFP) signaling was imaged using a laser scanning confocal microscope (TCS SP8, Leica, Solms, Germany), with an excitation wavelength of 488 nm and a 505–530 nm band-pass emission filter.

### 4.15. Co-Immunoprecipitation (Co-IP)

*N. benthamiana* leaves were infiltrated with 35S::CabZIP23-YFP^C^- MYC and 35S::CabZIP63-YFP^N^- HA, the leaves were harvested at 48 hpi and ground into a powder under liquid nitrogen. Total protein extracts were prepared using a protein extraction buffer (10% glycerol, 25 mM Tris-HCl, pH 7.5, 150 mM NaCl, 1 mM EDTA, 2% Triton X-100, 10 mM DTT, 1 × complete protease inhibitor cocktail (Sigma-Aldrich, St. Louis, MO, USA), and 2% (*w*/*v*) PVPP). The extracted proteins were incubated with monoclonal anti-MYC magnetic beads (Sigma-Aldrich) at 4 °C overnight. The beads were then collected with a magnet and washed 3 times with protein extraction buffer. Eluted proteins were separated by SDS-PAGE electrophoresis and immunoblotted using anti-HA-peroxidase antibody or anti-MYC-peroxidase antibody (Sigma-Aldrich). This experiment uses one sample for one biological repeat.

### 4.16. Quantitative Real-Time PCR (qRT-PCR) Assay

RT-qPCR was performed to detect the transcript levels of the selected genes as described previously by our group [83]. A Bio-Rad Real-Time PCR system and SYBR PreMix Ex Taq II system were used. CaActin (GQ339766) was used as an internal reference gene, and the data were analyzed by the Livak method. This experiment uses 4 samples for 4 biological repeats.

### 4.17. Statistical Analyses

Statistical analyses were performed with the DPS software package. Data are shown as means 6 SD obtained from three or four replicates; different letters indicate significant differences among means (*p* < 0.01), as calculated with Fisher’s protected least-significant-difference (LSD) test.

## Figures and Tables

**Figure 1 ijms-23-02656-f001:**
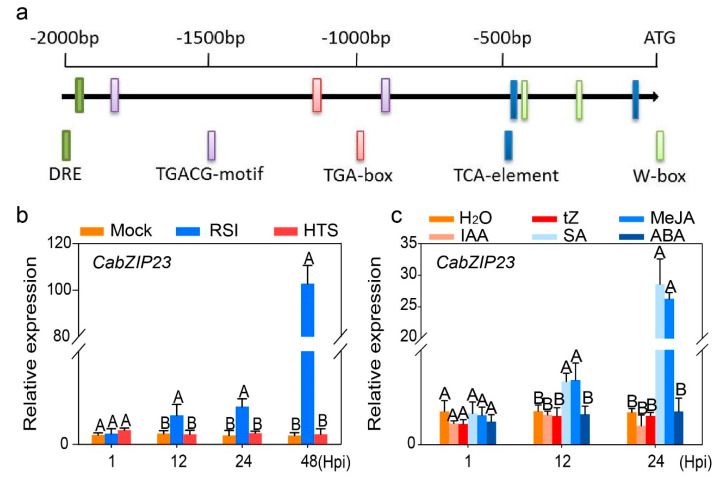
Analysis of cis-elements in CabZIP23 promoter and its relative transcription levels under different treatments. (**a**) Cis-elements in CabZIP23 promoter. (**b**) The relative transcriptional levels of CabZIP23 at 1, 12, 24 and 48 h after RSI or HTS by qRT-PCR. (**c**) The relative transcriptional levels of CabZIP23 at 1, 12 and 24 h after treatment of ddH_2_0, tZ, MeJA, IAA, SA, or ABA by qRT-PCR. Data presented are means of 6 standard errors (SE) of four replicates; different capital letters indicate significant differences among means (*p* < 0.01), as calculated with Fisher’s protected LSD test.

**Figure 2 ijms-23-02656-f002:**
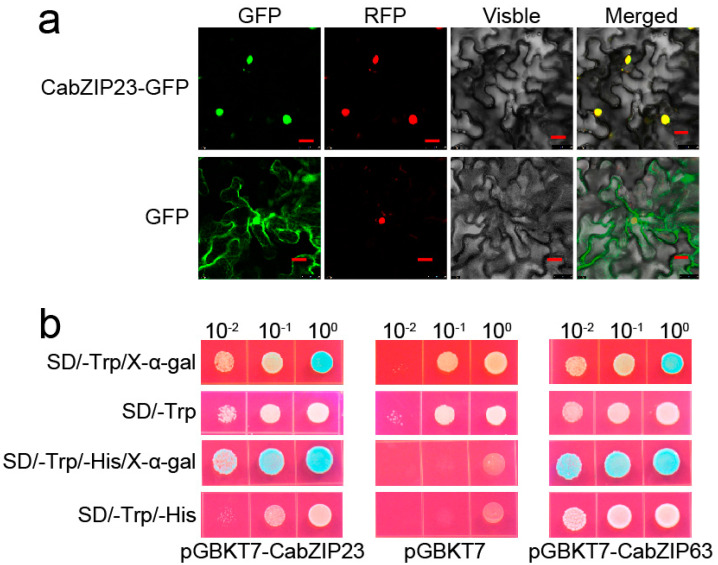
(**a**) The subcellular localization of CabZIP23 in epidermal cells of *N*. *benthamiana* leaves by agroinfiltration 25 μm. (**b**) Transcriptional activation of CabZIP23 by nutrition disfigurement assay in a. Bar, GAL4-based yeast two-hybrid system.

**Figure 3 ijms-23-02656-f003:**
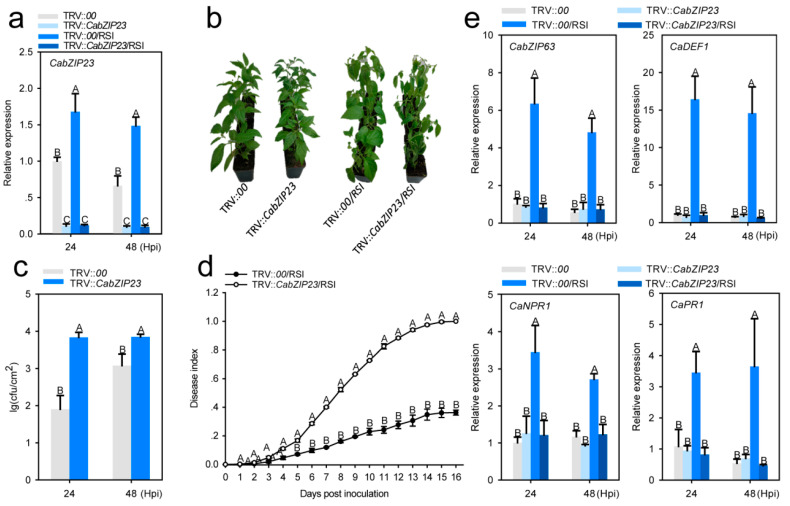
Effect of CabZIP23 silencing on the response of pepper plants to RSI. (**a**) qRT-PCR analysis of CabZIP23 silencing efficiency in TRV::CabZIP23 pepper plants. (**b**) The phenotype of the control and CabZIP23-silenced pepper plants upon RSI at 8 dpt. (**c**) Detection of *R. solanacearum* growth displayed with colony-forming units (cfu) in the leaves of the control and CabZIP23-silenced pepper plant leaves. (**d**) Disease index of *R. solanacearum* inoculated in the control and CabZIP23-silenced pepper plant leaves from 1 to 16 dpt. The values defined as: 0 (no symptoms), 1 (0–50% wilted leaves), 2 (25–50% wilted leaves), 3 (50–75% wilted leaves), and 4 (75–100% wilted leaves or death). (**e**) qRT-PCR analysis of the transcript levels of CabZIP63, CaDEF1, CaNPR1, and CaPR1 in the CabZIP23-silenced pepper plant leaves at 48 hpt. In (**a**,**c**,**e**), the data presented are means ± standard error (SE) of four replicates; different capital letters indicate significant differences among means (*p* < 0.01), as calculated with Fisher’s protected LSD test.

**Figure 4 ijms-23-02656-f004:**
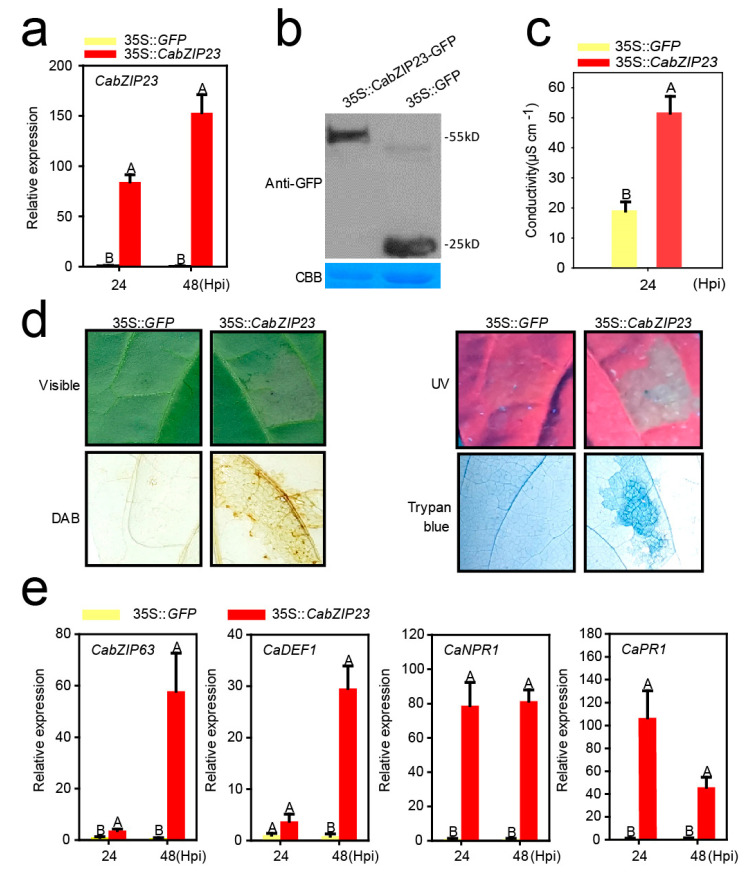
The transient overexpression of CabZIP23 triggers the hypersensitive response (cell death) and the expression of immune-related genes in pepper leaves. (**a**) The success of the transient overexpression of 35S::CabZIP23-GFP in pepper leaves detected by qRT-PCR at 24 and 48 hpi. (**b**) CabZIP23 expression in pepper leaves by transient overexpression of CabZIP23 by western blot assay at 48 hpi. α-GFP = anti-GFP antibody, CBB = Coomassie brilliant blue. (**c**) Quantitative measurement of electrolyte leakage caused by cell death using an ion conductivity meter. (**d**) Transient overexpression of CabZIP23 can cause cell death, and the hypersensitivity phenotypes were shown by staining under white light, UV light, DAB staining and Trypan staining, using 35S::GFP as a control. (**e**) The relative expression levels of defense-related genes including CabZIP63, CaDEF1, CaNPR1, and CaPR1 by qRT-PCR at 24 and 48 hpi in pepper leaves infiltrated with *A*. *tumefaciens* GV3101 cells harboring 35S::CabZIP23-GFP or 35S::GFP. In (**a**,**e**), the data presented are means ± SE of four replicates; different capital letters indicate significant differences among means (*p* < 0.01), as calculated with Fisher’s protected LSD test.

**Figure 5 ijms-23-02656-f005:**
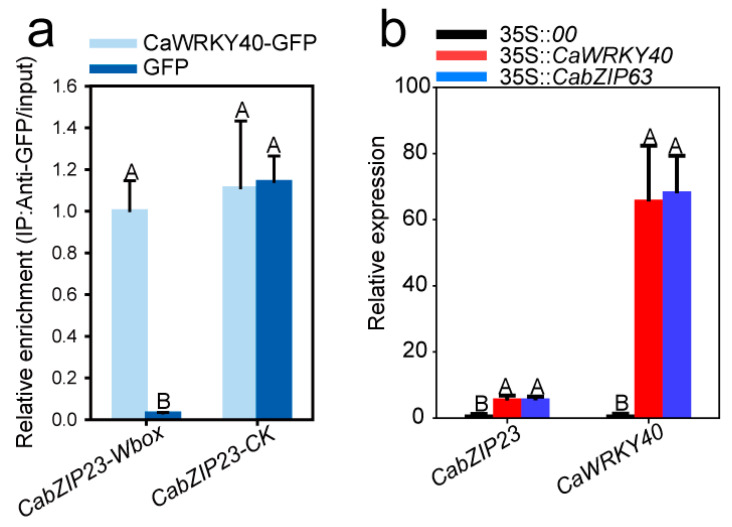
CabZIP23 was directly and positively regulated by CaWRKY40. (**a**) CaWRKY40 was enriched in the W-box containing promoter of CabZIP23 by ChIP-qPCR using specific primers and chromatins isolated from CaWRKY40 transiently overexpressed pepper leaves. (**b**) CabZIP23 was upregulated by the transient overexpression of CaWRKY40 or CabZIP63 in pepper leaves. The data presented are means ± SE of four replicates; different capital letters indicate significant differences among means (*p* < 0.01), as calculated with Fisher’s protected LSD test.

**Figure 6 ijms-23-02656-f006:**
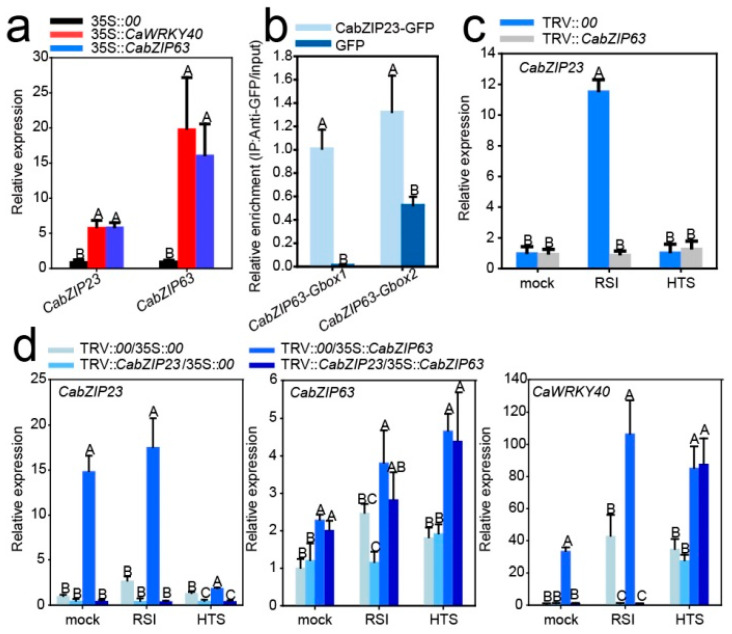
The effect of CabZIP23 silencing on the targeting and activation of CaWRKY40 by CabZIP63 by qRT-PCR. (**a**) The effect of CabZIP23 silencing on the binding of CabZIP63 to G-box1, G-box2 and C-box in the promoter of CaWRKY40 by ChIP-qPCR. (**b**)The effect of CabZIP23 silencing on the transcriptional activation of CaWRKY40 by CabZIP63 via qRT-PCR. (**c**) The expression level of CabZIP23 silencing on CabZIP63 under mock, RSI and HTS was detected by qRT-PCR. (**d**) CabZIP63 was overexpressed in silenced CabZIP23 plants and the expression levels of CabZIP23, CabZIP63 and CaWRKY40 under mock, RSI and HTS were detected by qRT-PCR. In (**a**,**c**,**d**), the data presented are means ± standard error (SE) of four replicates; different capital letters indicate significant differences among means (*p* < 0.01), as calculated with Fisher’s protected LSD test.

**Figure 7 ijms-23-02656-f007:**
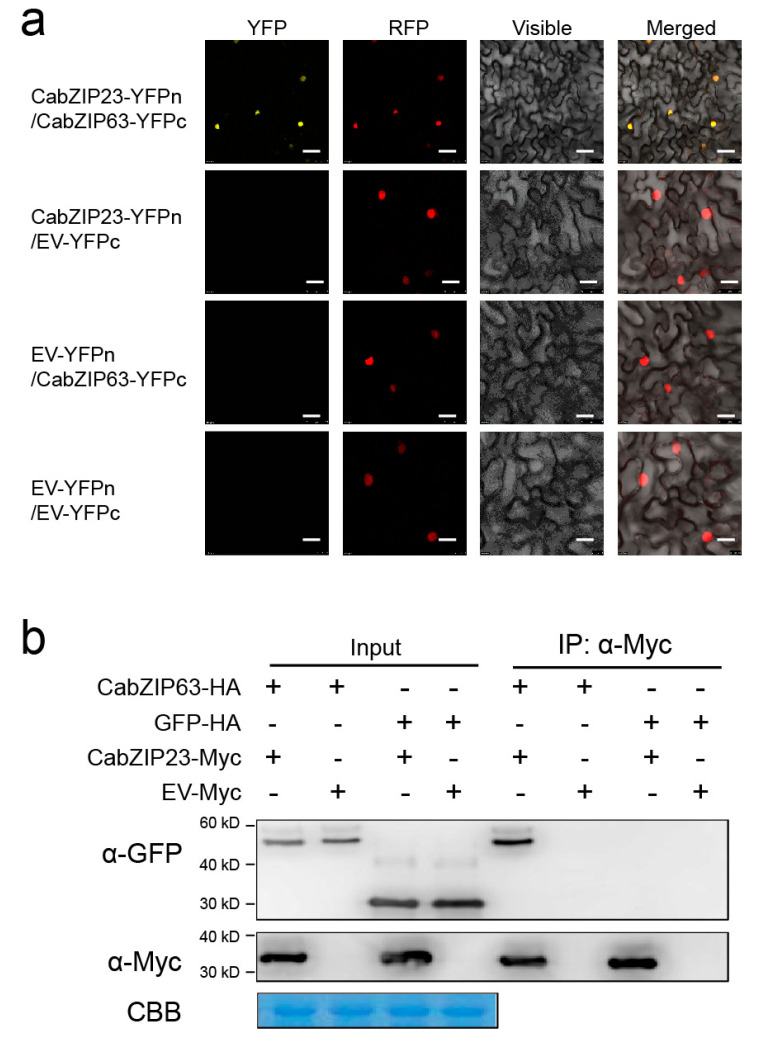
The interaction between CabZIP23 and CabZIP63. (**a**) The interaction between CabZIP23 and CabZIP63 by BiFC assay in *N. benthamiana* leaves infiltrated with Agrobacterium cells bearing CabZIP23-YFP^C^ and CabZIP63-YFP^N^ constructs. Bars = 25 um. (**b**) Interaction between CabZIP23 and CabZIP63 in vivo was analyzed by Co-IP experiment. Proteins were isolated from pepper leaves transiently overexpressing CabZIP23-Myc and CabZIP63-HA, and their interacting partners were immunoprecipitated with an antibody of Myc; the presence of CabZIP63 in the protein complex was assayed by western blotting using an antibody of HA.

**Figure 8 ijms-23-02656-f008:**
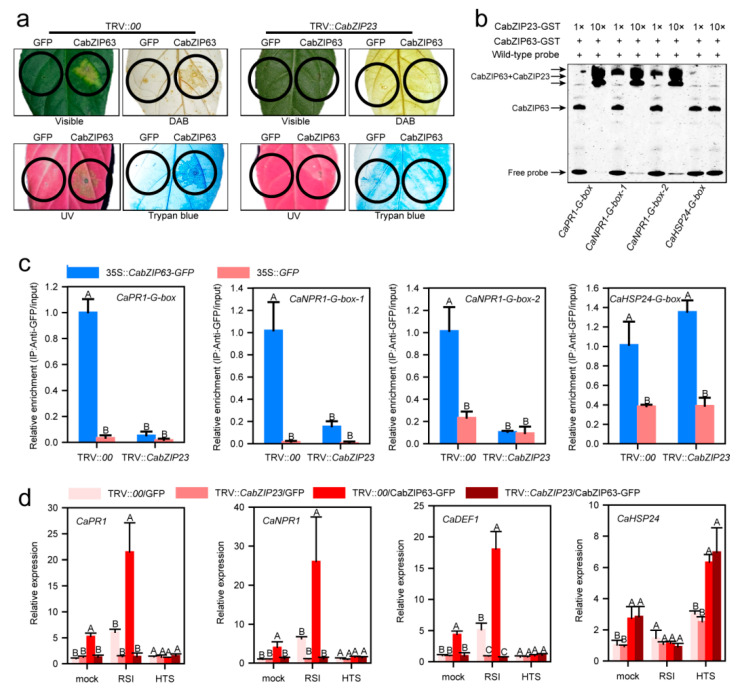
Effects of CabZIP23–CabZIP63 interaction on pepper immunity and thermotolerance. (**a**) HR mimicked cell death displayed by trypan staining, H_2_O_2_ accumulation displayed by DAB staining, and the upregulation of immunity related genes by the transient overexpression of CabZIP63 was significantly reduced by CabZIP23 silencing. (**b**) The application of excess CabZIP23 promoted the binding of CabZIP63 to the promoters of the tested immunity related genes, including CaNPR1 and CaPR1 but not CaHSP24, by an electrophoresis mobility shift assay (EMSA). (**c**) By ChIP-qPCR, CabZIP23 silencing reduced the binding of CabZIP63 to G-box, G-box1 and G-box2 of CaPR1 and CaNPR1, but did not affect the binding of CabZIP23 to the promoter of CaHSP24. (**d**) By qRT-PCR, the upregulation of CaPR1, CaNPR1 and CaDEF1 by the transient overexpression of CabZIP63 was reduced by CabZIP23 silencing, but the regulation of CaHSP24 by CabZIP63 was not affected by CabZIP23 silencing. In (**c**,**d**), the values are the means ± standard deviation from four biological replicates. Statistical analysis was performed by Fisher’s protected LSD test; different capital letters indicate significant difference at *p* < 0.01.

## Data Availability

Not applicable.

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
