# Peer review of "CabZIP23 Integrates in CabZIP63–CaWRKY40 Cascade and Turns CabZIP63 on Mounting Pepper Immunity against Ralstonia solanacearum via Physical Interaction"

_ijms, 2022, doi:10.3390/ijms23052656_

Round 1

Reviewer 1 Report

General features:

The general content of the article is correct. (The number of authors is high in relation to the content of the article).

1.- Statistical methods are not described in the materials and methods section. Its incorporation is recommended

2.- The writing of conclusions is recommended.

Author Response

The general content of the article is correct. (The number of authors is high in relation to the content of the article).

1.- Statistical methods are not described in the materials and methods section. Its incorporation is recommended

Response: Thank you very much for your good suggestion. We have added the statistical methods in the materials and methods section, please see line 503 to line 506 in the revised MS.

2.- The writing of conclusions is recommended.

Response: Thank you very much for your good suggestion. We have added the conclusions in the end of discussion, please see line 386 to line 389 in the revised MS.

Reviewer 2 Report

In this study, Qiaolin Lu et al. uncovered a molecular interaction cascade involved in pepper immunity against bacterium Ralstonia solanacearum. By using multiple stringent approaches including biochemistry, virus-induced gene silencing and genetic manipulation, authors demonstrate that CabZIP23 integrates with CabZIP63/CaWRKY40 and regulate the target gene expression in a positive feed-back loop. Interestingly, this molecular pathway seems not involved in the response to high temperature stress, highlighting the specificity of this regulation in bacteria defense. This discovery provides novel insights into pepper immunity against bacteria, which can be used in molecular breeding of future crops. The experiments are well designed and performed to a high standard. Manuscript is logically organized and data are solid and well presented. I don’t have any concerns and feel pleased to push this article to the next step to publication.  

Author Response

In this study, Qiaolin Lu et al. uncovered a molecular interaction cascade involved in pepper immunity against bacterium Ralstonia solanacearum. By using multiple stringent approaches including biochemistry, virus-induced gene silencing and genetic manipulation, authors demonstrate that CabZIP23 integrates with CabZIP63/CaWRKY40 and regulate the target gene expression in a positive feed-back loop. Interestingly, this molecular pathway seems not involved in the response to high temperature stress, highlighting the specificity of this regulation in bacteria defense. This discovery provides novel insights into pepper immunity against bacteria, which can be used in molecular breeding of future crops. The experiments are well designed and performed to a high standard. Manuscript is logically organized and data are solid and well presented. I don’t have any concerns and feel pleased to push this article to the next step to publication. 

Response: Thank you for your recognition of our MS.

Reviewer 3 Report

The main aim of the paper is the study the effect of CabZIP63 and CaWRKY40 in the resistance to high temperature stress and Ralstonia solanacearum in pepper

The paper shows an interesting information and a complete set of experiments. The material and methods include all the information, but the number of samples testes should be indicated. Practical implications should be indicated and discussed against other way to fight against RS.

Specific comments

Line. 71. Latin name of Pepper should be added the first time it appears in the text, and there are several species that refer to pepper so it should be stated to which or all are referring the authors.

Line 73. Latin name of every species should be stated.

Line 74. Authors of the name of every species should be stated. Ralstonia solanecaarum (Smith1896) Yabuuchi et al. 1996

Line 93. Reference needed.

Line 116. Different colours for H2O treatment should be used in figure 1C

Line 125.Why not in pepper?

Line 392. Practical implications? Discussion about other methodologies to fight against this disease.

Line 399. Name author Nicotiana name

Line 413.Characteristics of inbred line HN42?

Line 441. Number of samples and repetitions.

Line 450. Number of samples and repetitions. The same in the rest of the experiments

Author Response

The main aim of the paper is the study the effect of CabZIP63 and CaWRKY40 in the resistance to high temperature stress and Ralstonia solanacearum in pepper

The paper shows an interesting information and a complete set of experiments. The material and methods include all the information, but the number of samples testes should be indicated. Practical implications should be indicated and discussed against other way to fight against RS.

Specific comments

Line. 71. Latin name of Pepper should be added the first time it appears in the text, and there are several species that refer to pepper so it should be stated to which or all are referring the authors.

Response: Thank you very much for your good suggestion, we have added the latin name of pepper at the first time when it appears in the MS, please see line 69 in the revised MS.

Line 73. Latin name of every species should be stated.

Response: Thank you very much for your good suggestion, we have added the latin name of tomato or tobacco at the first time when it appears in the MS, please see line 72 in the revised MS.

Line 74. Authors of the name of every species should be stated. Ralstonia solanecaarum (Smith1896) Yabuuchi et al. 1996

Response: Thank you very much for your good suggestion, we have added the name of Ralstonia solanecaarum at line 74, please see line 74 in the revised MS.

Line 93. Reference needed.

Response: Thank you very much for your good suggestion, we have added the reference at line 93, please see line 93 in the revised MS.

Line 116. Different colors for H2O treatment should be used in figure 1C

Response: Thank you very much for your good suggestion, we have changed the color for H2O treatment in figure 1C, please see figure 1C in the revised MS.

Line 125.Why not in pepper?

Response: Thank you very much for this good question, Nicotiana benthamiana and pepper are closely related. In Nicotiana benthamiana leaves, the expression of fluorescent protein is higher, and the cell outline is clearer, so it is easier to observe.

Line 392. Practical implications? Discussion about other methodologies to fight against this disease.

Response: Thank you for your good suggestion, we added a practical implication of CabZIP23 and other shared signaling components at transcriptional level and regulator proteins at post-translational level as” thus to improve bacterial wilt resistance in pepper, these shared components at transcriptional level and specific regulators of post-translational regulation must be simultaneously and coordinately modified.” , please see Line 382 to Line 384.

Line 399. Name author Nicotiana name

Response: Thank you for your good suggestion, we have added the name author Nicotiana name, please see Line 393 in the revised MS.

Line 413.Characteristics of inbred line HN42?

Response: Thank you very much for this good question, HN42 is a pepper inbred line with middle level of thermotolerance and bacterial wilt resistance. We have added the characteristics in revised MS, please see line 392 to line 393 in the revised MS.

Line 441. Number of samples and repetitions.

Response: Thank you very much for your good suggestion, we usually use 4 samples for 4 biological repeats in transient overexpression in pepper leaves, please see line 434 in the revised MS.

Round 2

Reviewer 1 Report

Thank you very much for the update. The topic discussed is of great interest to me.

Kind regards,